# Fully Automatic Approach for Smoke Tracking Based on Deep Image Quality Enhancement and Adaptive Level Set Model

**Rimeh Daoudi** [1,2]**, Aymen Mouelhi** [1]**, Moez Bouchouicha** [2,]*****, Eric Moreau** [2] **and Mounir Sayadi** [1]

1 Université de Tunis, ENSIT, Laboratory SIME, Tunis 1008, Tunisia; rimehdaoudi@gmail.com (R.D.); aymen_mouelhi@yahoo.fr (A.M.); mounir.sayadi@ensit.u-tunis.tn (M.S.)
2 LIS-CNRS, Université de Toulon, Université Aix-Marseille, 83130 Toulon, France; eric.moreau@univ-tln.fr
* Correspondence: moez.bouchouicha@univ-tln.fr

**Abstract:** In recent decades, the need for advanced systems with good precision, low cost, and high-time response for wildfires and smoke detection and monitoring has become an absolute necessity. In this paper, we propose a novel, fast, and autonomous approach for denoising and tracking smoke in video sequences captured from a camera in motion. The proposed method is based mainly on two stages: the first one is a reconstruction and denoising path with a novel lightweight convolutional autoencoder architecture. The second stage is a specific scheme designated for smoke tracking, and it consists of the following: first, the foreground frames are extracted with the HSV color model and textural features of smoke; second, possible false detections of smoke regions are eliminated with image processing technique and last smoke contours detection is performed with an adaptive nonlinear level set. The obtained experimental results exposed in this paper show the potential of the proposed approach and prove its efficiency in smoke video denoising and tracking with a minimized number of false negative regions and good detection rates.

**Keywords:** convolutional autoencoder; smoke detection; image quality enhancement; deep learning models; level set models





## 1. Introduction

Wildfires are major natural disasters that can cause extensive damage to ecosystems and threaten human lives. It is an uncontrollable and destructive fire that rapidly spreads through vegetation, grasslands, or other flammable areas. Wildfires are typically triggered by a combination of factors, including the presence of abundant dry vegetation and favorable weather conditions like high temperatures, low humidity, and strong winds. The sources of ignition for wildfires are diverse and can range from natural causes like lightning strikes to human activities such as campfires, careless disposal of cigarettes, or even intentional acts of arson. Besides the destructive nature of wildfires, the smoke from wildfires can have severe human health risks and environmental consequences as it can contribute to air quality degradation, disrupt the balance of ecosystems, and even impact the behavior and survival of wildlife.

Early fire and smoke detection are crucial; some recent methods and tools focus on smoke and fire detectors [1] due to their low cost. However, they have some limitations, such as poor performance in open and wide areas and the detection time, which is dependent on the distance between the detector and the smoke source [2]. Recent monitoring systems such as remote sensing technologies, satellite imagery, and ground-based sensors rely on artificial intelligence-based algorithms in detecting and monitoring the outbreak of fires and their spread. In those systems, vision software runs at its core and automatically detects the presence of an event.

In the last years, like other fields, computer vision-based fire and smoke detection methods have attracted the attention of researchers [3–6]. Compared to conventional

methods [7–11], they have many advantages like the large area coverage, the real-time response, and the high accuracy.

Some of the existing approaches integrated various combinations of hand-crafted traditional techniques to detect smoke image features like shape, color, and texture [12–14]. At the same time, others exploit the moving nature of smoke to extract the motion characteristics [13,15,16] (motion value, direction, energy, convex hull), achieving good results considering some particular cases.

In recent years, deep learning-based algorithms, especially CNNs [17–21], have gained so much attention, and many researchers investigated those models and achieved good precision results. Hence, the use of those methods is still challenging since they rely on supervised learning and require huge databases for training, and their effectiveness depends on various factors, including the strong hardware systems for implementation and the quality and diversity of the dataset used for training. A database is one of the bases of a deep learning model, and the provided results largely depend on the comprehensiveness and the authenticity of training data. Industries and institutions involved in deep learning spend a lot of time and energy collecting datasets. The significant investment required to acquire these datasets adds value to deep learning training and underscores their importance. The availability of public smoke and fire databases with their ground truth labels is one of the biggest challenges related to deep learning models. In addition, smoke and wildfire images captured from fixed cameras, drones in motion, or other imaging devices are often affected by noise, artifacts, or other imperfections that can affect their interpretation and analysis. To address these challenges, Image processing techniques, such as noise reduction, image enhancement, and image restoration, are employed to improve the quality of acquired images, making them more suitable for interpretation.

In contrast to these previous methods and related challenges, a further solution is described in this paper. In this study, we propose a fully automatic hybrid method designated for noise removal and image quality improvement followed by smoke tracking in video sequences. The proposed benchmark consists of two stages. In the first stage, we construct an image denoising and quality enhancement path based on lightweight convolutional encoder-decoder new architecture. In the second stage, we propose a novel scheme for smoke tracking in videos provided by cameras in motion. The proposed scheme consists of tracking the evolution of smoke that appears in different shapes and from different angles in consecutive frames. First, a decision function to indicate the appearance of smoke in the first frames in a recorded video based on hand-crafted features like energy and color. The goal of those primary steps is to extract the foreground image that contains possibly smoke region candidates. Second, smoke motion tracking with a nonlinear adaptive level set framework. In this work, our main contributions are three folds: (i) we develop a novel fully unsupervised pipeline for smoke tracking in videos captured from a source camera in motion, (ii) we analyze the convolutional encoder-decoder structures, and we suggest a novel view of its architecture that is designated especially to reconstruct and denoise wildfire images, and (iii) we track the smoke regions evolution starting from the small appearance regions in the first video frames. Besides giving good detection results, the advantage of the proposed method is to be fully unsupervised, which avoids the need for labeled, big, and high-resolution databases. In addition, in theory, it is apt for deployment in real-world situations with low computational cost.

Extensive experiments and comparisons with the state-of-the-art methods confirm the good performance of the proposed benchmark. In addition, our method is able to detect the smoke at early stages with a small appearance from a moving camera, which, to our best knowledge, has not been discussed in related research.

The rest of the paper is organized as follows: in Section 2, we provide an overview of recent methods related to fire and smoke detection; Section 3 exposes the proposed approach; Section 4 discusses the obtained results and evaluation of our method; and in Section 5, we end up with conclusions and future works.

## 2. Related Works

Computer vision and deep neural networks have emerged as powerful strategies for studying the behavior of fire and smoke. Numerous methods have been introduced in the literature to leverage these technologies. Kaabi et al. [22] introduced a method for smoke detection. First, a motion-based feature extraction with the GMM algorithm was employed; then, a trained DBN classifier based on the feature of smoke was used to detect the smoke region in videos. Xu et al. [23] proposed an end-to-end method for smoke detection based on a deep saliency network; the framework was used to extract the smoke saliency map via a pixel level and object level salient CNN. Yifan et al. [24] proposed a real-time detector network for fire and smoke based on a light YOLOv4. The model is based mainly on three modules: the MobileNetv3, the BiFPN, and a new feature extraction technique with depthwise separable convolution and attention block. The method attends good performance with a minimum of trainable parameters. Cao et al. [25] proposed a bidirectional LSTM network for forest fire and smoke detection in videos, and it consists of bidirectional learning of the discriminative spatiotemporal features. Ali Khan et al. [26] exploit the advantages of transfer learning to train the VGG19 model to localize wildfires; then, they build a network composed of unmanned aerial vehicles communicated to an assistance center to simplify and increase the data transmission process. Frizzi et al. [27] proposed a new CNN architecture for fire and smoke segmentation and classification. The network is composed of coding and decoding paths and achieves better accuracy compared to similar methods with low false positives (clouds, haze) and in optimal segmentation time. Aymen et al. [28] integrate the non-linear adaptive level set method with an artificial neural network model to track the forest fire regions in wildfire videos; the method consists of estimating and localizing the possible fire contours by analyzing the chromatic and statistical features with the linear discriminant analysis combined with an artificial neural network. Then, a level-set algorithm was applied to refine the segmentation results. The method gives good results in terms of speed and accuracy. An interesting approach proposed by [29] concentrates and studies the architecture and the training of networks; it combines the self-attention mechanism with a multi-scale feature connection for real-time fire and smoke detection. The authors first fused the feature maps of the network into a radial connection; then, they applied a permutation attention mechanism to gather the relevant information, and designed a feature fusion block to increase the detection efficiency. The method gives good results compared to standard proposed methods.

Recent technologies based on advanced artificial neural network architecture achieve good results, but they still have limitations, especially when trying to implement them for daily use because of the need for huge databases for training and the strong/expensive computation resources. From this, many methods attempt to detect fire and smoke by building a pipeline integrating the image processing techniques and by exploiting the smoke textural features (Table 1). Ref. [30] built a statistical model combined with an optical flow algorithm for real-time fire and smoke detection. The method first extracts the smoke and fire-like regions with frame differential steps, a color model of fire and smoke, and a foreground accumulation technique. A motion feature discriminating model with the optical flow was applied to the first resulting image to extract the final fire and smoke regions. In [31], the fire presence decision in video frames is reached by analyzing the color variations and periodic behavior of the flame with the temporal and spatial wavelet transform algorithm. Ref. [32] proposed a fire detection system consisting of modeling the color information in the CIE *L\*a\*b* color space and detecting the motion of fire pixels with a background subtraction technique.

**Table 1.** Overview of main methods and findings in the smoke detection literature.

| | Color Space | Features | Method | Accuracy (%) |
|---|---|---|---|---|
| Hashemzadeh [7] | RGB | Motion | CNN SVM | 97.6 |
| Pundir [33] | RGB | Motion Texture Color | Deep CNN | 97.4 |
| Yin [34] | RGB | Motion | CNN | 97.0 |
| Toreyin [35] | YUV | Motion Energy Disorder | Wavelet transform | - |

## 3. The Proposed Smoke Tracking Method

The proposed approach presented in Figure 1 is based mainly on two stages: the first stage is a reconstruction path, which aims to enhance the image quality and remove noise, and the second stage consists of smoke detection and tracking using a new algorithm based on smoke image features extraction steps combined with the nonlinear adaptive level set method.

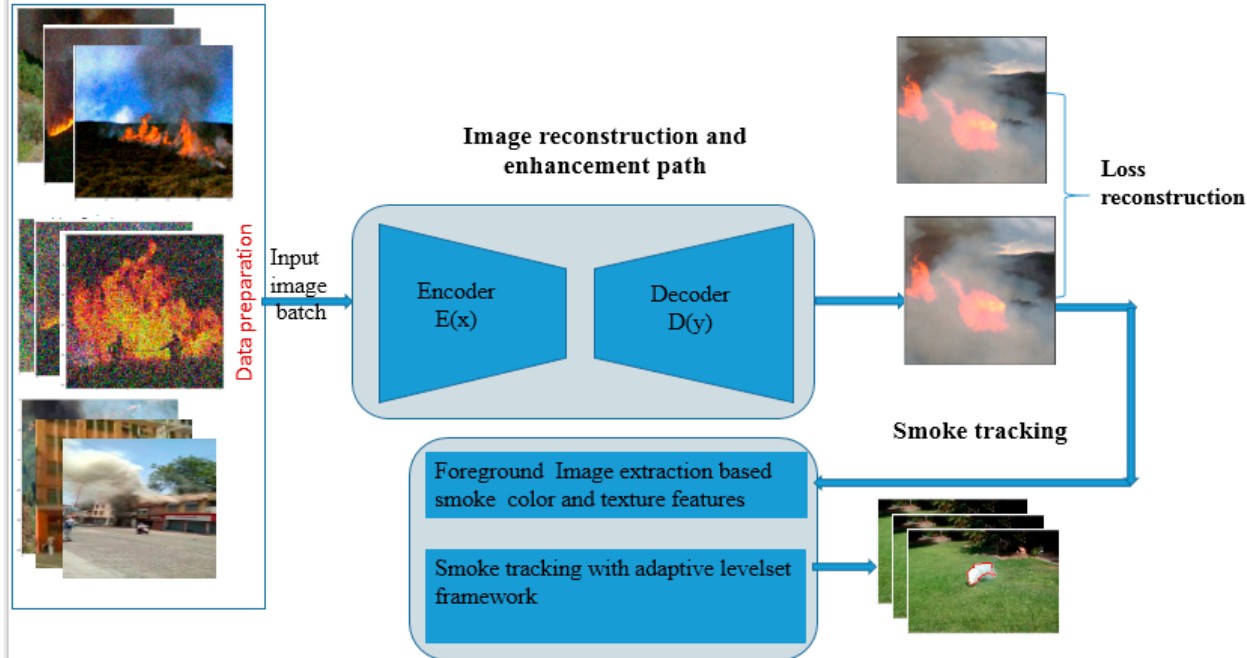

**Figure 1.** Flowchart of the proposed method.

### 3.1. Image Enhancement Path

3.1.1. Dataset Preparation

In this work, to train and validate the proposed convolutional autoencoder CAE architecture, we used a dataset collected by our laboratory research team. The dataset contains 450 RGB images, and it was divided into 80% used for training and 20% for validation of the network. Note that to optimize the network learning, the dataset used to train the network contains a diverse combination of clean and noisy images with varying noise levels, as shown in Figure 2. Subsequently, every inputted image was normalized and resized to a consistent size ($256 \times 256 \times 3$).

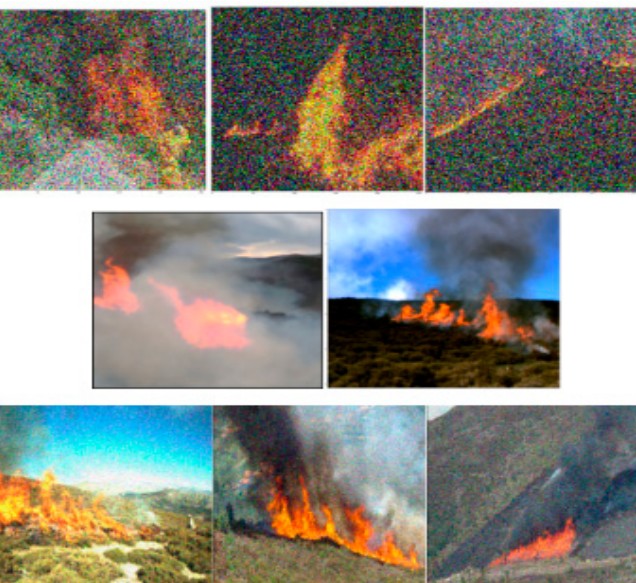

**Figure 2.** Frames from the training database. Row (1): noisy images with Gaussian noise: $\mu = 0$, $\sigma = 1$, $\rho = 0.7$. Row (2): clean original images. Row (3): noisy images with $\mu = 0$, $\sigma = 1$, $\rho = 0.1$.

### 3.1.2. The Proposed Lightweight CAE Network Architecture

Deep learning dimensionality reduction methods such as Variational auto-encoders and their variants and generative adversarial networks have gained so much attention. Some methods exploit the obtained latent space representation to help classification and detection models better learn the distribution of information to achieve high-precision results, while others tend to generate new data, for example, the data understanding approaches.

Many feature fusion and selection techniques employ neural architecture for better cross-scale feature network topology. Auto-encoders [36] have demonstrated their efficacy in mapping and modeling the data and proved their performance in data mapping. They have been successfully applied in various tasks, including data manipulation, image generation, and reconstruction.

Convolutional auto-encoders (CAE) [37] build upon the concept of auto-encoders by incorporating a consecutive stacked convolutional layers, CAE is designed to learn a specialized representation of image data, it involves mapping an input image data $x$ to a latent space representation h via an Encoder $h = E(x)$ and then reconstruct it to obtain the output image $x$ via a decoder D with $x = D(h)$.

To improve the learning process and help the model predict the missing values in a corrupted image $\widetilde{x}$ with noise, we employed a data mixing technique, which involved injecting a batch of data that was corrupted with varying noise levels into the model. It is important to note that this task is completely unsupervised, meaning that labeled data are not required to train and fine-tune the CAE model.

Let us represent the feature representation of $\widetilde{x}$ as $h$ by denoting it as:

$$h = \sigma(w_\vartheta \widetilde{x} + b_\vartheta) \tag{1}$$

where $\widetilde{x}$ is the input image to be reconstructed.

The reconstructed image $z$ is defined by:

$$z = \delta(W_k h + b_k) \tag{2}$$

where $W_v, b_v, W_h, b_h$, are the weights and biases of the CAE model.

Our designed CAE architecture operates by first flattening the input image data into a feature representation vector. It incorporates a sliding window and multiple hidden layers consisting of convolution, pooling, and batch normalization operations. This architecture

enables the network to learn distinctive contextual patterns and global and local features from various regions within the image.

Figure 3 illustrates the detailed architecture of the proposed CAE model. It consists of two paths: the encoding path begins with two convolution layers with a kernel size of 3 × 3 and 64 filters. This is followed by a normalization layer, a max-pooling operation, and ends with a 3 × 3 convolution layer followed by a max-pooling operation.

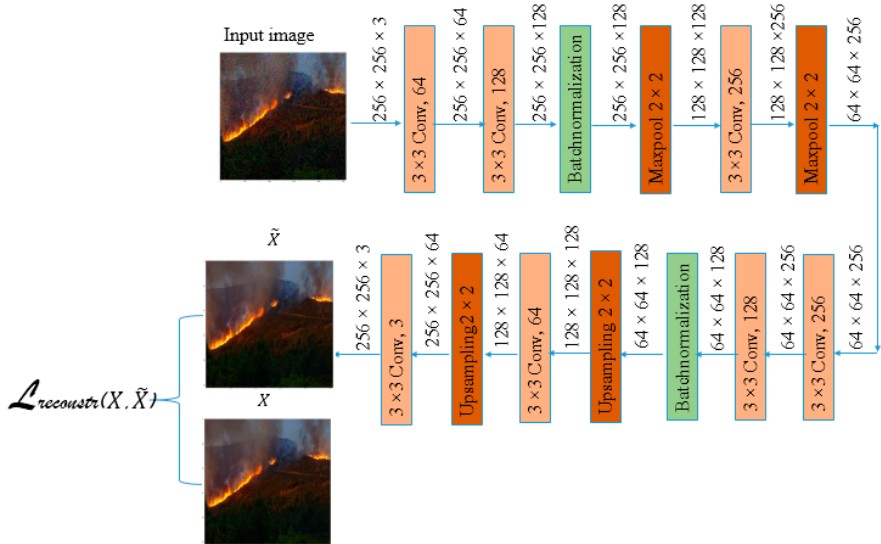

**Figure 3.** An overview of the proposed architecture of the CAE network.

The decoding path of our model consists of two convolution operations. The first operation utilizes 256 kernels, while the second operation employs 128 kernels. These blocks are followed by a batch-normalization and up-sampling operations. Next, the feature map is passed through a convolution block and another up-sampling operation. The decoding path concludes with a convolution layer.

The architecture we propose is designed to be lightweight and promotes local connectivity between neurons across different layers. This enables a hierarchical decomposition of fire and smoke images, representing low and high-level features at varying scales and levels. Additionally, it effectively blurs the input image data and maintains fine details like contours and corners.

Note that the model was trained and validated on images for computation and time simplicity reasons. After training the network and saving the best-achieved weights and biases, the video smoke sequences are decomposed into frames, and then those frames are fed one by one into the network for quality improvement and noise removal.

### 3.2. The Tracking Path

#### 3.2.1. Color Modeling for Smoke Detection

Our aim in this study is to detect the smoke regions in video sequences acquired from a camera with motion. The video dataset used for testing the tracking algorithm is available at http://signal.ee.bilkent.edu.tr/VisiFire/Demo/SampleClips.html (accessed on 12 April 2023). The smoke in the chosen videos is characterized by its variable shape, appearance, and motion.

The smoke usually appears with grayish colors [38], either dark or light levels of gray. The appearance of smoke depends on the combustion process. In this work, we consider the light smoke since we want to find out the smoke at earlier stages. In terms of computation, the RGB color model has less complexity than other models, but in smoke color recognition, the extraction of smoke color-related information in RGB domain space is not suitable. In this paper, the HSV color domain is used, which is a more people-oriented

information model. Our experimental results show that to obtain the primary foreground image, every selected HSV smoke pixel should satisfy the following conditions:

$$\text{(i)} \quad 0.26 \leq H \leq 0.53 \tag{3}$$

$$\text{(ii)} \quad 0.01 \leq S \leq 0.13 \tag{4}$$

$$\text{(iii)} \quad 0.7 \leq V \leq 1 \tag{5}$$

Foreground thresholded images may contain both smoke and disturbances, which are to be differentiated in the following phases.

The pixels that satisfy the above color decision rule are set as smoke region candidates/foreground images, and the rest are the background. Afterward, the foreground image underwent a refinement process that involved removing regions with small areas. Additionally, the smoke region candidates were smoothed using a morphological closing operation, and any holes within the regions were filled. These pre-processing steps significantly improved the quality of the obtained mask and reduced the number of false detections.

### 3.2.2. Smoke Feature Extraction

To enhance the threshold result of the color model, we need to ensure that the grayish detected region is a smoke region. Many methods exploit the motion feature by using a differential method, but since we are detecting smoke regions from a moving camera, the subtraction frames technique is not efficient. We studied the different image features between consecutive frames, such as correlation, energy, mean, entropy, homogeneity, and contrast, etc., and we found that in a video, the energy and the homogeneity of an image increase with the presence of smoke. So, to consider a region that appears during consecutive frames as a smoke region, the $\alpha$ and $\beta$ terms must satisfy the conditions:

$$\alpha(I_K, I_{K-1}) > 0 \tag{6}$$

$$\beta(I_K, I_{K-1}) > 0 \tag{7}$$

where $\alpha$ is the difference value of the energy feature between the current frame $I_K$ and the previous frame $I_{K-1}$ two consecutive frames and $\beta$ is the difference value of the homogeneity feature between the current frame $I_K$ and the previous frame $I_{K-1}$.

### 3.2.3. Level Set Segmentation

To proceed with the smoke image quality enhancement results, we propose in this paper to adapt a nonlinear adaptive level set for smoke tracking refinement in video sequences. The level set-based method is an extension of the active contour without edges model proposed by Chan–Vese [39]. The simplified Chan–Vese model gives an approximation to an image in a piecewise constant piece; then, the same authors designed a penalty term by measuring the distance between the approximation and the original image, which helps the model be robust with noise. The model has been extended into many methods, e.g., the vector-valued images for color images [40], the multiphase level set [41], Tsai et al. [42] gave a piecewise smooth approximation for image segmentation and denoising derived from [39]. Ref. [43] proposed a maximum posterior (MAP) function to model the external energy. These methods are robust against the noise since they exploit the regional image features. However, they are inefficient in segmenting images with inhomogeneous regions due to the ignoring of local image features.

The nonlinear adaptive level set function presented by [44] evolves an initial curve by minimizing an energy function. The evolution of the curve takes into consideration local and global region features.

Let $\Phi$ be the level set function of an inputted gray level image defined on image domain $\Omega$ with $C = \{(x, y) \mid \Phi(x, y) = 0\}$ the zero-level set at the boundary region(C), $\Phi > 0$ inside the contour C and $\Phi < 0$ outside the contour region C.

The formulation of the energy function to be minimized for a gray-level image is as follows:

$$E_g(\Phi) = \mu \int_\Omega \frac{1}{2}(|\nabla\Phi| - 1)^2 \, dxdy + \lambda \int_\Omega \delta_\varepsilon(\Phi)g|\nabla\Phi| \, dxdy + \nu \int_\Omega g \, H_\varepsilon(-\Phi)dxdy \quad (8)$$

where the three terms of the right-hand side of (8) are, respectively, the penalization term, which controls the property of the signed distance during the evolution process of C, the weighted length of the curve, and the weighted area inside $\Phi$. $\mu$, $\lambda$ and $\nu$ are the parameters controlling the effects of the three terms of the energy function.

Large $\mu$ and small $\lambda$ help the algorithm to filter high-frequency noise and detect objects with finer details. $H(.)$ is a regularized Heaviside function, and its derivative is the regularized Dirac function $\delta(.)$. $g$ is an edge detector function.

The evolution direction function takes into account the global and local statistical features and is computed with a posterior probability obtained by the Bayesian rule. The conditional probability is obtained with a Gaussian function with color channels of an RGB image. The direction evolution function was determined as follows:

$$Adf(x,y) = \frac{1}{3} \sum_{j=1}^{3} \frac{(I_j(x,y) - \mu_{1j})^2}{2\sigma_{1j}^2} - \frac{1}{3} \sum_{j=1}^{3} \frac{(I_j(x,y) - \mu_{2j})^2}{2\sigma_{2j}^2} - \ln\left(\frac{1}{3} \sum_{j=1}^{3} \frac{\sigma_{1j}}{\sigma_{2j}}\right) \quad (9)$$

where $j = 1, 2, 3$ are the RGB channels, respectively, of a processed frame.

To increase the evolution speed of the level set function in the homogeneous regions and decrease it when crossing the real boundaries, a velocity parameter was adapted by [44]:

$$v(x,y) = 2\left[\frac{1}{1 + \exp(-\zeta \, Adf(x,y))} - 0.5\right] \quad (10)$$

where $\zeta$ is a constant parameter that controls the nonlinear degree of the velocity. This parameter is computed by estimating the absolute difference between the current frame $I_k$ and the previously processed frame $I_{k-1}$. Each iteration is updated using the smoke motion feature to speed up the level set evolution [28].

The identification of the smoke regions is difficult since the model can not distinguish between smoke regions and clouds, and the task became more difficult in images with haze or fog or in daytime captured images. To overcome this challenge, it is crucial to accurately well estimate and automatically initialize the parameters to ameliorate the segmentation process accuracy. The proposed scheme contributes by ameliorating the obtained masks and decreasing the number of false negatives ROIs.

This first stage generates a binary mask for the candidate smoke area. This mask is used for the level set algorithm initialization. The adaptive level set proved his performance in detecting the smoke borders efficiently in different frames and segmenting the image regions into smoke and background. Hence, this model requires a good parameter estimation and a precise initialization of the level set functions. The parameters of the model are determined with an empirical assessment. An overview of the scheme is presented in Figure 4.

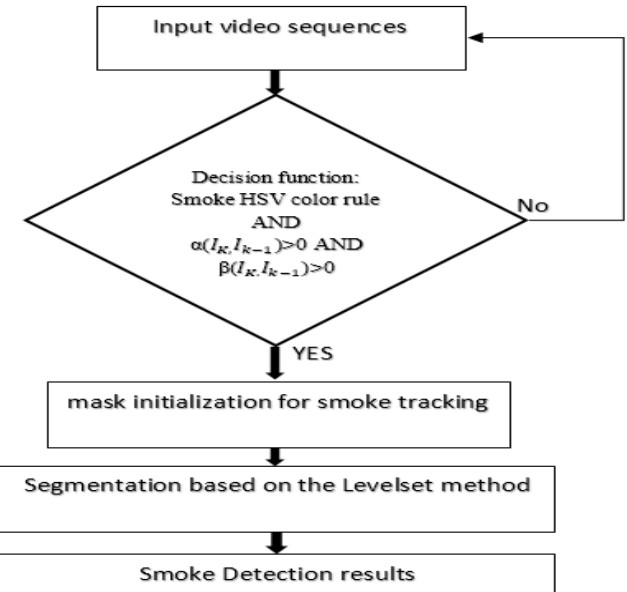

**Figure 4.** Workflow of the proposed smoke detection stage in video sequences.

## 4. Experimental Results and Discussion

*4.1. CAE Configuration and Image Quality Enhancement Results Evaluation*

The training and validation of the CAE model was implemented using Python programming language and with keras and tensorflow backend. Fixing and fine-tuning the hyperparameters enables us to have a robust network with good results. We tested several values to fix the parameters and opted to set the learning rate to 0.001. The trained model outputted a reconstructed video image with some quality improvement. Then, the reconstructed video image was fed to the segmentation stage to detect smoke regions.

Table 2 exposes different parameter configurations.

**Table 2.** Parameters configuration of the CAE model.

| Input Dimension | (256, 256, 3) |
|---|---|
| Epochs number | 130 |
| Loss | Formulation combines the SSIM, MSE, MAE |
| Weight decay | $1 \times 10^{-5}$ |
| Optimizer | Adam |
| Trainable parameters | 299,139 |

To optimize the learning process of the CAE network, a loss formulation introduced by Roy et al. (2021) [45] has been integrated. It is a weighted sum of three losses: the MSE loss, the MAE loss, and the SSIM loss.

The equation of reconstruction loss is defined by:

$$l = \alpha \times l_{MSE} + \beta \times l_{SSIM} + \gamma \times l_{MAE} \tag{11}$$

where $\alpha$, $\beta$, $\gamma$ are hyper-parameters determined by accurate testing $\alpha = \beta = \gamma = 0.5$ [19].

The SSIM loss is defined by:

$$l_{SSIM}(x_{ij}, \widetilde{x}_{ij}) = 1 - SSIM(x_{ij}, \widetilde{x}_{ij}) \tag{12}$$

where the formulation of the structural similarity index measure is:

$$SSIM(x, \widetilde{x}) = \left[ l(x, \widetilde{x})^{\alpha} \cdot c(x, \widetilde{x})^{\beta} s(x, \widetilde{x})^{\gamma} \right] \tag{13}$$

where $l(.)$, $c(.)$, and $s(.)$ are the luminance, contrast, and structural comparison functions, respectively.

The loss function plays a crucial role in image generation and manipulation problems as it guides the model in optimizing its weights and provides insight into its performance in data modeling. Various approaches have been proposed in the literature, often relying on the mean squared error (MSE) loss for evaluation. In our approach, we employed both losses, the reconstruction loss, and the MSE loss, and we found that the reconstruction loss yielded the best performance. As depicted in Figure 5, the model achieved its lowest loss value of 0.05 at the 120th epoch. Notably, the proposed architecture of the convolutional autoencoder (CAE) yielded excellent results even with a small dataset and without data augmentation techniques.

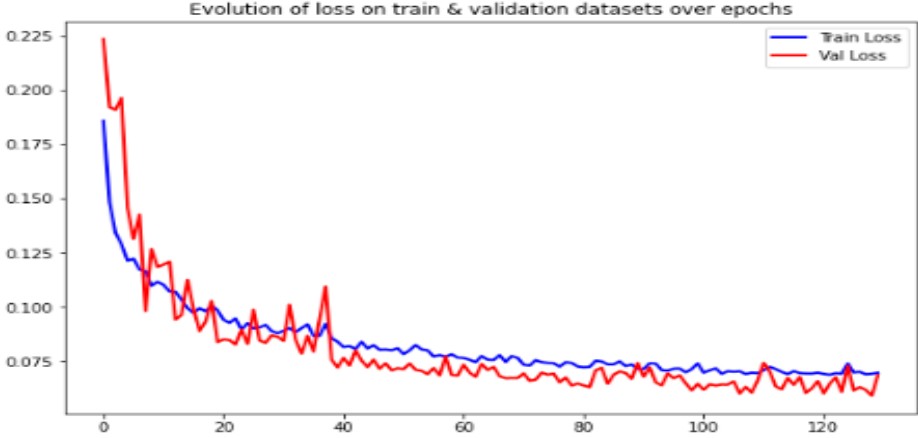

**Figure 5.** The reconstruction loss evolution curves.

In order to objectively evaluate the proposed CAE model in image denoising and reconstruction, SSIM, PSNR, and MSE evaluation criteria were employed.

The outcomes of the experiment conducted on a randomly selected image from the test samples are provided in Table 3. Additionally, Table 4 displays the denoising results achieved using our method in comparison to various conventional filters and a technique proposed in the literature. The results undeniably illustrate the superiority of the CAE over other denoising methods. Our CAE model excels in eliminating noise points while simultaneously retaining fine details and enhancing the overall image resolution. These findings serve as compelling evidence of the effectiveness and efficiency of our proposed approach in denoising images, highlighting its potential for practical applications. The enhanced image quality resulting from our approach will greatly facilitate the analysis of the reconstructed video images in the subsequent stages. It is important to note that our dataset comprises both clean images and noisy images with varying levels of noise. As a result, our proposed CAE is capable of performing both image reconstruction and denoising tasks simultaneously. This dual functionality enables the CAE to effectively address the challenges posed by noise in the dataset and produce high-quality, noise-free images for further investigation and analysis.

**Table 3.** Quantitative results of the CAE network.

| Image | PSNR | SSIM | MSE | Processing Time(s) |
|---|---|---|---|---|
| Filtered with the proposed CAE | 73.2 | 0.91 | 0.0030 | 0.6 |
| Filtered with a median filter | 72.3 | 0.79 | 0.0040 | 0.04 |
| Noisy image | 69.04 | 0.33 | 0.0081 | - |

**Table 4.** Comparative evaluation results of the proposed CAE architecture with different filters and methods using the SSIM evaluation criterion.

| Filtering Type | SSIM |
|:---:|:---:|
| Noisy | 0.33 |
| Median filter | 0.79 |
| Gaussian filter | 0.90 |
| Gondara [46] | 0.89 |
| The proposed CAE | 0.91 |

In Figure 6, a set of reconstructed resulting images obtained from the designed CAE architecture is displayed. Through visual evaluation, we observed a slight improvement in image contrast. The CAE demonstrated its capability to effectively eliminate artifacts in noisy images and accurately reconstruct the fire and smoke regions, as well as other parts of the images.

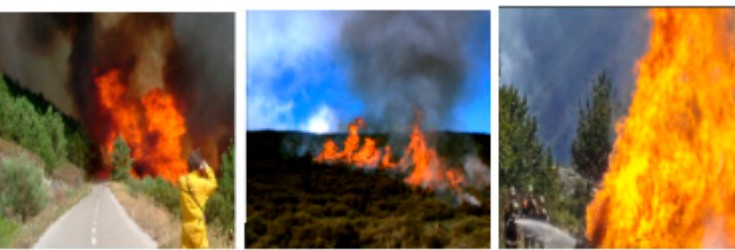

**Figure 6.** Construction results of the fire and smoke images from the test samples with the proposed lightweight CAE architecture.

### 4.2. Video Smoke Tracking Path

In this section, we provide detection results evaluation of the reconstructed smoke video images using the proposed approach, which involves foreground extraction followed by the adaptive level set algorithm. Figure 7 illustrates a selection of sample frames from the video test set. These video frames were passed through the CAE model to eliminate noise, reduce image size while preserving relevant features, and enhance the overall video image quality. Subsequently, the samples were preprocessed to extract the initial mask of the smoke region candidates. This mask was further refined using an adaptive level set model to obtain the final detection of smoke regions in each frame.

A subjective evaluation can be observed from Table 5, which shows that the proposed method successfully separated the smoke area from the background image. To verify the effectiveness of the method, we compared the segmentation results with those obtained using Fuzzy c-means (FCM), K-means, and spatial fuzzy c-means combined with the level-set algorithm (Spatial FCM). Both our method and the three algorithms used for comparison were able to segment the smoke area. However, only the proposed algorithm yielded enhanced segmentation results with a minimum of false detections. The clustering algorithms used were unable to differentiate between regions with the same texture features. For example, in images (a), (c), and (d), our benchmark was able to differentiate between different regions with the same appearance and extract the smoke area from the complex background image. It is worth noting that to identify the weaknesses of our method, Table 5 showcases some critical particular cases from the test dataset with low accuracy in detection. The results are promising and demonstrate the superiority of our method.

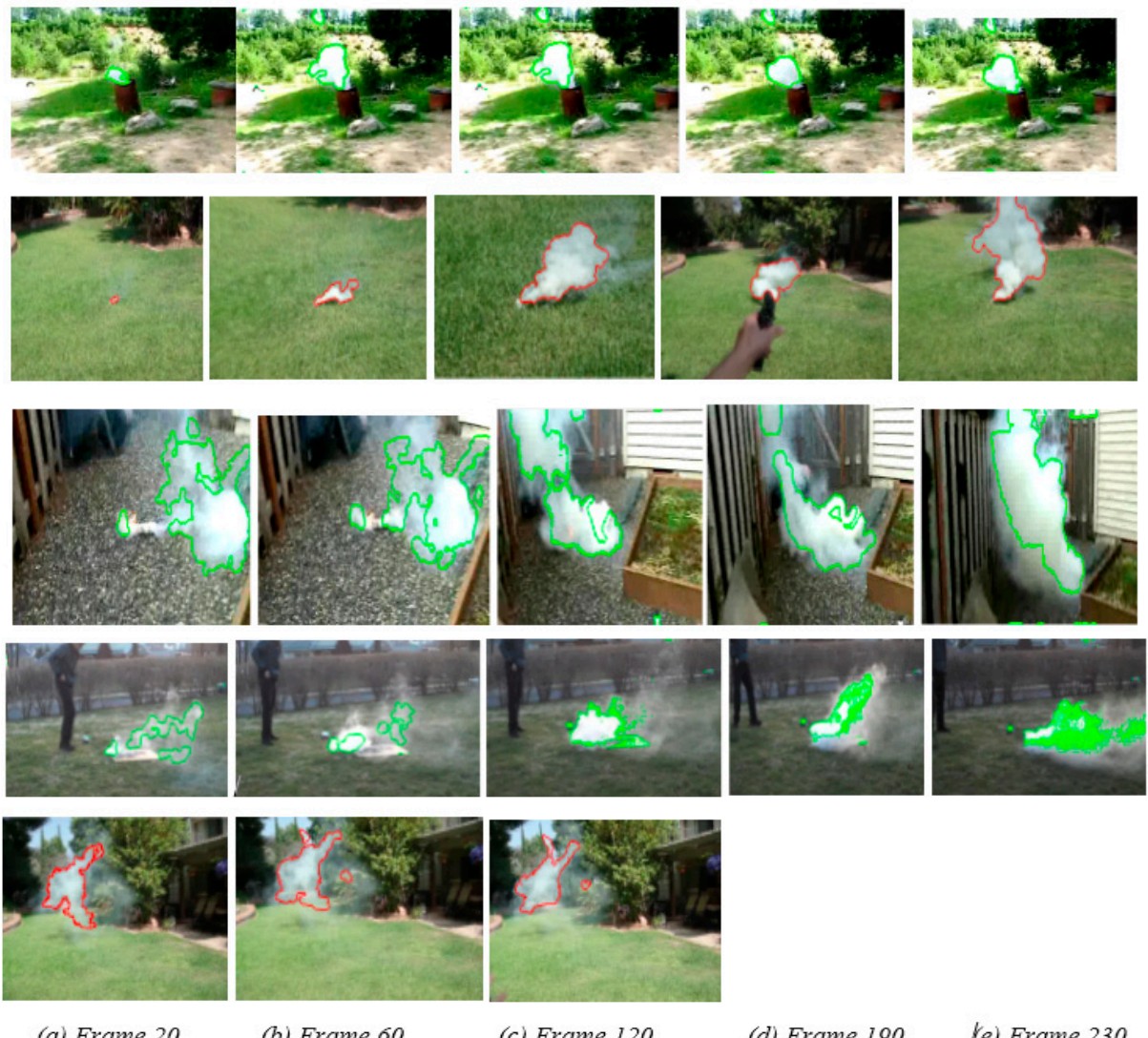

(a) Frame 20    (b) Frame 60    (c) Frame 120    (d) Frame 190    (e) Frame 230

**Figure 7.** Smoke tracking results with the proposed method with illustration of subsequent frames extracted from five different videos from row (1) to row (5): final smoke boundaries detection results in (**a**) Frame 20, (**b**) Frame 60, (**c**) Frame 120, (**d**) Frame 190, and (**e**) Frame 230.

However, the edges of segmented regions are still not smooth enough, and the reason for this is that the model still confuses clouds with cluttered skies. This is somewhat justified because we use a smoke color model in our preprocessing steps, and the choice of the smoke color model depends on the dataset used in this work (nearby smoke and a moving camera in an environmental scene). Another reason for the low accuracy is that smoke, depending on its combustion level and density, can exhibit an array of grayish color shades, including the darkest ones, which can affect the segmentation results of our algorithm.

We believe that the strength of the proposed method lies in being a fully unsupervised benchmark by integrating a new workflow for smoke denoising and tracking, resulting in very promising segmentation accuracy and detection time. Moreover, it is accessible for employment in real-world situations.

**Table 5.** Visual comparison of segmentation results with different algorithms.

| Original Image | Ground Truth | Our Method | FCM | Spatial FCM [47] | K-means |
|---|---|---|---|---|---|
| (a) | | | | | |
| (b) | | | | | |
| (c) | | | | | |
| (d) | | | | | |
| (e) | | | | | |
| (f) | | | | | |

In order to illustrate quantitatively the effectiveness of the proposed benchmark, we selected 24 image frames from 7 smoke videos. These frames were chosen to represent various smoke locations under different weather conditions and situations. We calculated the Jaccard similarity index and the Dice coefficient between the segmentation results and the manually annotated images. The experimental results for the test set, including the mean, maximum, and minimum evaluation criterion values, are reported in Table 6. In addition, Table 7 exposes a quantitative evaluation of the proposed method on four videos from the test set. $F_{tot}$ is the total number of frames in a video sequence, $F_s$ is the number of frames containing smoke regions in a video, $F_t$ is the number of frames detected as smoke with the proposed method, $R_d$ is the detection rate of a video. The detection rate formula is defined as follows:

$$R_d = \frac{F_c}{F_t} \tag{14}$$

where $F_c$ is the number of frames that are correctly classified with the proposed method and $F_t$ is the number of frames in a video sequence.

**Table 6.** Quantitative evaluation of the smoke segmentation using the adaptive level set-based algorithm in terms of accuracy and processing time.

|  | Max | Min | Mean |
|---|---|---|---|
| Jaccard index (%) | 92.1 | 80.5 | 90.1 |
| Dice coefficient (%) | 90.0 | 79.6 | 89.5 |
| Processing time/frame (s) | 5.3 | 4.9 | 5.0 |
| Processing time/video (s) | 484.9 | 235.4 | 376.3 |

**Table 7.** Experimental results of smoke detection in videos using the proposed scheme.

| Video Sequences | $F_{tot}$ | $F_s$ | $F_t$ | $R_d(\%)$ |
|---|---|---|---|---|
| Video 1 | 675 | 674 | 670 | 99.20 |
| Video 2 | 407 | 403 | 403 | 99.02 |
| Video 3 | 360 | 360 | 360 | 100.00 |
| Video 4 | 310 | 308 | 308 | 99.35 |
| Video 5 | 150 | 150 | 150 | 100.00 |

To proceed with the evaluation process in our experiments, we evaluated the efficiency of the CAE architecture, and we compared the segmentation accuracy values obtained with the proposed pipeline with the segmentation accuracy obtained with the traditional filtering method, which is the median filter and the segmentation accuracy obtained by applying the level set method on noisy images. Table 8 reports measurements performed on 24 image frames in terms of Jaccard similarity index and dice coefficient. Based on these experiments, we can deduce that the proposed pipeline provides encouraging segmentation results by reaching more than 90% accuracy and effectively identifies smoke regions within video frames, accomplishing this task within a reasonable processing time, particularly when the regions are uniform and exhibit statistical differences from the background. The purpose of employing this approach is to eliminate the need for manual parameter configuration and enhance the segmentation process with the level set method.

**Table 8.** Performance evaluation of the smoke segmentation pipeline compared to other segmentation methods in terms of accuracy segmentation and processing time.

|  | Jaccard Similarity Index (%) | | Dice Coefficient (%) | | Detection Time/ Image(s) | |
|---|---|---|---|---|---|---|
|  | Min | Max | Min | Max | Min | Max |
| segmentation of raw images | 70.6 | 89.9 | 73.2 | 89.8 | 8.2 | 9 |
| segmentation of denoised images with a median filter | 64.7 | 87.3 | 65.5 | 86.9 | 4.6 | 5.1 |
| segmentation of denoised images with the proposed pipeline | 80.5 | 92.1 | 79.6 | 90.0 | 4.9 | 5.3 |

The experimental results show that those steps were able to help the level set function track the evolution of the smoke borders in a video sequence in an optimal time and with a minimum of false detections.

The detection of smoke is highly important in practical scenarios, particularly when dealing with fire and smoke in wildfire video images. Both qualitative and quantitative observations clearly demonstrate that the model performs well in identifying wildfire smoke in reconstructed video images, especially in detecting the appearance of small

smoke regions in the first frames. However, the lower accuracy segmentation values in particular frames are justified by the similarity between intensity regions. Due to the appearance of smoke in various color combinations and different spatial domains, making it challenging to identify using traditional conventional methods. Another contributing factor to the lower accuracy values is the manual segmentation process. In certain cases, it can be difficult to determine whether a pixel belongs to the fire, smoke, or background class. For instance, when smoke is present against a cloudy sky or when flames are obscured by smoke and segmented as background, the model detected it because of its characteristics. While the resulting segmentation is not really false, it can influence the model's performance evaluation.

## 5. Conclusions

Efforts to improve early detection and response to wildfires and smoke continue to evolve with advancements in technology and the use of data-driven approaches. By understanding the behavior and characteristics of smoke, it becomes possible to develop more effective strategies to prevent and mitigate their devastating effects. In this work, we introduced a new lightweight convolutional autoencoder architecture and incorporated an automatic scheme based on color and texture features characteristics of smoke and a nonlinear adaptive level set algorithm. Also, we provided a detailed explanation of the implementation and training process, as well as the experimental results and evaluation for both the reconstruction and segmentation stages in video sequences. Our proposed workflow yielded visually and quantitatively satisfactory results, effectively improving image quality, eliminating noise, and accurately detecting smoke boundaries in RGB video frames.

In future works, our objective is to conduct a thorough investigation into feature extraction and selection techniques, as well as generative models, in order to generate high-resolution and large-scale databases. Also, extend this methodology to track fires and smoke in more complex scenarios. Additionally, we plan to adapt this approach to detect fires and smoke with small appearances, as they serve as the first clue of a wildfire recipient, and implement the algorithm in an embedded system to use it for real-world applications. Ultimately, we intend to generalize our method and reproduce our findings with a diversified data pool.

**Author Contributions:** R.D. conceived the presented approach. R.D., M.B. and A.M. developed the theory and performed the computations. M.S., M.B. and A.M. verified the analytical methods and were involved in planning and supervising the work. M.S. and E.M. encouraged R.D. to design, and they supervised the findings of this work. All authors discussed the results and contributed to the final manuscript. All authors have read and agreed to the published version of the manuscript.

**Funding:** This research received no external funding.

**Data Availability Statement:** Experimental video data covered in this paper are available at the following http://signal.ee.bilkent.edu.tr/VisiFire/Demo/SampleClips.html accessed on 12 April 2023. For the image dataset provided by our laboratory research please contact the authors by email.

**Conflicts of Interest:** The authors declare no conflict of interest.

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
