# Peer review of "Fully Automatic Approach for Smoke Tracking Based on Deep Image Quality Enhancement and Adaptive Level Set Model"

_electronics, doi:10.3390/electronics12183888_

Round 1

Reviewer 1 Report

1. The Introduction section requires a comprehensive revision to incorporate a review of more recently published works in the field. By including recent research findings, the authors can establish the context of their study and highlight the novel aspects that differentiate their work from previous contributions.

2. The level of English needs to be improved. Several instances of grammatical errors and typographical mistakes are observed throughout the text. Additionally, some sentences are unclear and difficult to understand. Therefore, it is recommended that the authors carefully review the entire manuscript and rectify all these typographical and grammatical issues. Doing so will significantly enhance the clarity and overall quality of the paper.

Significant improvement is needed. 

Author Response

Dear Professor,

Thank you for your helpful comments. We have revised our paper accordingly and feel that your comments helped clarify and improve our paper. Please find our response  to reviewer’s specific comments  below.

  • The Introduction section requires a comprehensive revision to incorporate a review of more recently published works in the field. By including recent research findings, the authors can establish the context of their study and highlight the novel aspects that differentiate their work from previous contributions

The aim of this work is to create an unsupervised benchmark for denoising and reconstructing the acquired images and tracking the wildfire smoke in video sequences in order to help specialists detect a smoke recipient and make objective and fast intervention. Recently, multiple works are focused on the supervised approaches and machine learning methods which need a huge labeled high resolution dataset for training. Furthermore, many supervised methods who relies on neural networks are usually applied with sophisticated learning algorithms and parametrized to be trained depending on the smoke features in order to optimize the architecture of the segmentation algorithm and enhance the training process in terms of computational time and detection performance.  

So, in order to address the implementation complexities and the challenges of these supervised segmentation strategies and improve performances of smoke tracking in acquired video sequences, an unsupervised smoke tracking pipeline, based on a convolutional autoencoder architecture and a baseline algorithm combines the merits of image processing techniques and nonlinear adaptive level set method, has been performed in this study.

The introduction has been edited and include more recent works related to fire and smoke detection, also it has changed as suggested to improve clarity.

  • The level of English needs to be improved. Several instances of grammatical errors and typographical mistakes are observed throughout the text. Additionally, some sentences are unclear and difficult to understand. Therefore, it is recommended that the authors carefully review the entire manuscript and rectify all these typographical and grammatical issues. Doing so will significantly enhance the clarity and overall quality of the paper.

English level of the paper was carefully reviewed

Reviewer 2 Report

General Comments: The subject addressed in this article, "Fully Automatic Approach for Smoke Tracking Based on Deep Image Quality Enhancement and Adaptive Level Set Model" is worthy of investigation. The authors propose a method to deal with the problem of wildfire detection (smoke) through computer vision techniques and deep learning.

Strengths:

• A good methodology was used by the authors to deal with the problem of smoke tracking by using a deep learning approach.

• A good revision of the state-of-the-art was done.

All prediction methods were clearly presented.

• The description of the state-of-the-art was well-done.

Weaknesses:

• The authors must extend the discussion about the strategy to detect wildfires by tracking smoke because it is assumed a priory that smoke detection is considered the best way to detect fire.

• It is always better to work with real images acquired from real scenarios. Thus, the first step that added noise to images is not good. Why did the authors want to create a complex image instead of getting focus on the real ones?

• The dataset is a little bit confusing due to the lack of ground truth information about the detection. Thus more experimental results are required.

Author Response

Dear Professor,

Thank you for your helpful comments. We have revised our paper accordingly and feel that your comments helped clarify and improve our paper. Please find our response to reviewer’s specific comments below.

  • The authors must extend the discussion about the strategy to detect wildfires by tracking smoke because it is assumed a priory that smoke detection is considered the best way to detect fire.

Smoke detection is a good way to detect wildfires in early stages, which make the fires controllable and help firefighters assessing it. The aim of this paper is to track the evolution of smoke regions in a video sequences. In future work we aim to extend this approach to build a full system for fire and smoke detection implemented in an unmanned aerial vehicle and able to send an alarm to an assistance center to prevent specialists.

  • It is always better to work with real images acquired from real scenarios. Thus, the first step that added noise to images is not good. Why did the authors want to create a complex image instead of getting focus on the real ones?

All available open source images and videos databases are already preprocessed, high resolution databases, who are built to be used for training neural networks architectures. The lack of real scenarios images acquired under different weather conditions and presents artefacts conducts us to prepare a dataset for simulation. The dataset used for training and evaluating the proposed CAE network is collected by our laboratory research team, since we are trying to build a model for image quality enhancement and noise removal, we need to train the network with noised image data that’s why we added noise to our dataset, and to match the real world captured images we added different levels of noise.

  • The dataset is a little bit confusing due to the lack of ground truth information about the detection. Thus more experimental results are required.

This has been changed in text

We just want to point out on the numerical evaluation of the tracking results, semantic segmentation of a set of frames extracted from the smoke video data is evaluated using Jaccard similarity index and dice coefficient. In fact, to obtain a precise evaluation of the proposed method in terms of segmentation accuracy and computational time, we selected 24 frames from 7 smoke sequences and compared them with manually segmented frames. Also, to proceed on the quantitative measurements, we evaluated the detection rate from 4 smoke videos containing smoke regions with different location and shapes by computing the detection rate per video and evaluating the number of true detections with the proposed method. Besides the encouraging obtained results, we believe that the efficiency of the proposed method appears in the low cost and its feasibility of implementation in real-time surveillance devices.

Reviewer 3 Report

The Autoencoder is used for Denoise. And levelest Algo is used for segmenting the smokes.

Concerns:

1. The paper is lacking experiments to show the effectiveness of the pipeline. For example: segmentation accuracy from autoencoder output vs. segmentation accuracy from raw image vs. segmentation accuracy from other denoise methods. 

2. Why not directly use segmentation methods to segment the smokes? 

3. Do the paper use the same setting across all variations? e.g. black smoke vs white smoke? 

Author Response

Dear Professor,

Thank you for your helpful comments. We have revised our paper accordingly and feel that your comments helped clarify and improve our paper. Please find our response to reviewer’s specific comments below.

  • The paper is lacking experiments to show the effectiveness of the pipeline. For example: segmentation accuracy from autoencoder output vs. segmentation accuracy from raw image vs segmentation accuracy from other denoise methods.

We are updated our paper, Table 7 was added to show the effectiveness of the pipeline, it reports the measurements of the three values suggested by you: the segmentation accuracy from autoencoder (proposed method), segmentation accuracy from raw image and segmentation accuracy from median filter. Experiments shown encouraging results of the proposed pipeline and proves the efficiency of the CAE proposed model in noise removal and image reconstruction.

  • Why not directly use segmentation methods to segment the smokes?

the CAE new architecture is integrated to enhance and maintain the detection results of acquired images even when implemented on a hardware system.

  • Does the paper use the same setting across all variations? e.g. black smoke vs white smoke?

The proposed pipeline that we built is designated for tracking smoke with grayish colour levels, the detection of smoke with other variations (e.g: black smoke) is possible with simple modification in the proposed colour smoke model (described in the paper), the choice of smoke colour variation indicated in our paper related to availability of the video dataset that we have.

Round 2

Reviewer 1 Report

The authors adequately addressed all my concerns, making this manuscript suitable for acceptance in its current form. 

Author Response

-

Reviewer 2 Report

Dear Authors, 

thank you for considering some of the suggestions. Even if several changes appear in the newer version of the manuscript some significant experimental results are required in order to accept the proposed method. The strategy is clear but the experimental section is too weak and it is not possible for me to validate the contribution. 

Finally, I strongly recommend doing more experiments and adding more results and discussion about it. 

Best Regards, 

Reviewer!

Author Response

Response: In this new submission, more experimental results was added, also we extend the discussion and evaluate the limits of our method. In table 5 we provide a set of images with complex background and we report the segmentation results of some clustering methods, which they are unsupervised, including the proposed method against manual segmentation provided by specialists. In the paragraph just right after the table, a discussion of ablation experiments was added. A subjective evaluation shows a promising results given by our method compared to the comparison algorithms (FCM, Fuzzy FCM integrated with levelset and Kmeans), Unlike those algorithms, it is able to differentiate efficiency between areas with the same texture features and extract only the smoke region even in complex similar background.

Reviewer 3 Report

With added experiments and data, the authors demonstrated the effectiveness of using an encoder-decoder network to denoise the input images help enhance the features for smoke detection. The proposed frame work has shown large improvement compare to using traditional vision denoise approaches such as medium filter. 

Author Response

-

Round 3

Reviewer 2 Report

Summary: The authors made significant changes in the Experimental Results Section. Thus, the paper is in an improved version. Considering the above, I recommend publishing the manuscript as it appears.